# Computer-Selected Antiviral Compounds: Assessing In Vitro Efficacies against Rift Valley Fever Virus

**DOI:** 10.3390/v16010088

**Published:** 2024-01-05

**Authors:** Cigdem Alkan, Terrence O’Brien, Victor Kenyon, Tetsuro Ikegami

**Affiliations:** 1Department of Pathology, The University of Texas Medical Branch at Galveston, Galveston, TX 77555, USA; cialkany@utmb.edu; 2Discovery Chemistry, Genentech, Inc., South San Francisco, CA 94080, USA; obrien.terry@gene.com; 3Atomwise Inc., San Francisco, CA 94103, USA; victor@atomwise.com; 4Sealy Institute for Vaccine Sciences, The University of Texas Medical Branch at Galveston, Galveston, TX 77555, USA; 5Institute for Human Infections and Immunity, The University of Texas Medical Branch at Galveston, Galveston, TX 77555, USA

**Keywords:** Rift Valley fever, Arumowot virus, Heartland virus, Dabie bandavirus, Gc fusion loop, small molecule compound, AI-based computational drug discovery

## Abstract

Rift Valley fever is a zoonotic viral disease transmitted by mosquitoes, impacting both humans and livestock. Currently, there are no approved vaccines or antiviral treatments for humans. This study aimed to evaluate the in vitro efficacy of chemical compounds targeting the Gc fusion mechanism. These compounds were identified through virtual screening of millions of commercially available small molecules using a structure-based artificial intelligence bioactivity predictor. In our experiments, a pretreatment with small molecule compounds revealed that 3 out of 94 selected compounds effectively inhibited the replication of the Rift Valley fever virus MP-12 strain in Vero cells. As anticipated, these compounds did not impede viral RNA replication when administered three hours after infection. However, significant inhibition of viral RNA replication occurred upon viral entry when cells were pretreated with these small molecules. Furthermore, these compounds exhibited significant inhibition against Arumowot virus, another phlebovirus, while showing no antiviral effects on tick-borne bandaviruses. Our study validates AI-based virtual high throughput screening as a rational approach for identifying effective antiviral candidates for Rift Valley fever virus and other bunyaviruses.

## 1. Introduction

Rift Valley fever (RVF) is a zoonotic viral disease caused by the Rift Valley fever virus (RVFV), a member of the *Phlebovirus* genus within the *Phenuiviridae* family. It is transmitted by mosquitoes and is endemic to sub-Saharan African countries, Egypt, Madagascar, the Comoros, Saudi Arabia, and Yemen [1]. RVF is characterized by a high incidence of abortions, fetal demise, fetal malformations, and a high lethality rate among newborn animals within livestock animals, including sheep, cattle, goats, and camels. Individuals can contract the RVF through the bite of an infected mosquito or through close contact with the bodily fluids of infected animals. While the majority of RVF patients experience a self-limiting febrile illness, some may progress to more severe conditions such as lethal hemorrhagic fever, encephalitis, or retinitis, potentially resulting in partial or complete blindness [2]. The RNA genome of RVFV consists of three segments: large (L)-, medium (M)-, and small (S)-segments. The L-segment encodes the viral RNA-dependent RNA polymerase, the M-segment encodes the envelope proteins Gn and Gc, the 78 kD protein, and the nonstructural M (NSm) protein. Lastly, the S-segment encodes the nucleoprotein (N) and the nonstructural S (NSs) protein.

Recognizing its significance to public health and animal health, RVF is classified as a Category A Priority Pathogen by the National Institute for Allergy and Infectious Diseases (NIAID). It also falls under the select agent category of both the U.S. Department of Health and Human Services (HHS) and Agriculture (USDA) in the United States. Furthermore, RVFV is listed as a priority disease on the World Health Organization’s (WHO) R&D Blueprint, and it is a notifiable disease to the World Organization for Animal Health (OIE).

At present, a limited number of licensed live-attenuated or inactivated veterinary vaccines for RVF are accessible to endemic countries [3,4]. Nonetheless, the challenge in eradicating or preventing RVF outbreaks persists, primarily due to the widespread presence of infected mosquitoes in endemic regions and the difficulty in achieving comprehensive vaccine coverage for susceptible livestock animals. Nonendemic countries are also preparing for potential introductions of RVF by investing the development of effective countermeasures, including vaccines and antivirals. Currently, there are no licensed RVF vaccines or antivirals available for human use.

Many small molecules have been identified as potent inhibitors of RVFV, each acting through distinct mechanisms [5]. These include nucleoside analogs (Ribavirin, Favipiravir, BCX4430, 6-azauridine) [6,7,8]; an oxidizer of viral lipid membranes (LJ001) [9]; an inhibitor of viral RNA replication and egress (Sorafenib) [10]; inhibitors of viral entry and/or RNA synthesis (E225-0969, E528-0039, G118-0778, G544-0735) [11]; inhibitor of post viral entry and/or RNA replication (1-N-(2-(biphenyl-4-yloxy)ethyl)propane-1,3-diamine and the derivatives) [12]; a disruptor of N-RNA binding (Suramin) [13]; an mTOR inhibitor (Rapamycin) [14]; an autophagy enhancer (SMER28) [15]; an antioxidant (NSC62914) [16]; HSP90 inhibitors (17AAG, BAPT A-AM) [17]; a Protein Phosphatase-1 inhibitor (1E7-03) [18]; a cell membrane modulator affecting viral fusion (25-hydroxycholesterol) [19]; a ubiquitin proteasome inhibitor (Bortezomib) [20]; and inhibitors with unknown mechanisms (FGI-106, 5,6-dimethoxyindan-1-one analog, mitoxantrone, toosendanin) [6,21,22,23].

The RVFV Gc protein, a class II membrane fusion protein similar to alphavirus E1 and flavivirus E proteins [24], presents an attractive target for antiviral development. Rational design of small molecules necessitates structural insights into viral fusion proteins. Under normal pH conditions, the RVFV Gc protein is impeded by the Gn protein. However, exposure to low pH in the late endosome induces rearrangement of the Gn-Gc glycoprotein complex. This rearrangement facilitates the insertion of a hydrophobic fusion loop into the outer leaflet of the host membrane, followed by stem zippering to form a fusion pore connecting viral content to the host cytoplasm [25]. Although the RVFV-6, a 19-amino acid-long Gc stem peptide, has been reported to potentially disrupt the interaction of domain III with the trimer core during stem zippering [26], no small molecules specifically interrupting the RVFV Gc fusion process have been reported. The X-ray structure of RVFV Gc in the post-fusion form has been elucidated [27]. Thus, this study employed screening of millions of commercially available small molecules to identify binders of RVFV Gc using an Artificial Intelligence (AI)-based virtual high throughput screen (vHTS). Subsequently, the in vitro efficacy of chemical compounds targeting the Gc fusion mechanism was evaluated. This AI-assisted screening successfully identified three potent small molecules that target the RVFV Gc fusion mechanism.

## 2. Materials and Methods

### 2.1. Media, Cells, and Viruses

Vero cells (ATCC CCL-81) were maintained at 37 °C in DMEM (Gibco, Thermo Fisher Scientific Inc., Waltham, MA, USA), containing 10% fetal bovine serum, penicillin (100 U/mL, Gibco), and streptomycin (100 µg/mL, Gibco), in a humidified cell culture incubator with 5% CO_2_. The recombinant Rift Valley fever virus MP-12 vaccine strain (rMP-12) was recovered from Vero cells via reverse genetics as described previously [28]. Wild-type Arumowot virus Ar 1286–64 strain (AMTV), wild-type Heartland virus MO-4 strain (HRTV), and wild-type Dabie Bandavirus HB29 strain (SFTSV: formerly Severe Fever with Thrombocytopenia Syndrome virus) were obtained from the World Reference Center for Emerging Viruses and Arboviruses (WRCEVA) at The University of Texas Medical Branch at Galveston (UTMB). Cells and viruses used in this study were verified to be mycoplasma free at the University of Texas Medical Branch at Galveston (UTMB) Next Generation Sequencing Core Facility.

### 2.2. Screening of Small Molecule Compounds

To identify small molecule binders of RVFV Gc, a vHTS was performed using the AtomNet^®^ platform (Atomwise, Inc., San Francisco, CA, USA), as described previously [29,30,31,32,33,34,35]. A library of 8,062,379 small molecule with drug-like properties (Mcule v20191203) was screened and ranked using the post-fusion RVFV Gc protein (PDB ID: 6EGT, W821H mutant) [27]. This binding site was defined by structural groove formed by C823, G824, C825, F826, N827, R776, C777, H778, L779, V780, D961, and P1135.

The top ranked compounds were reduced to a select 94 compounds with drug-like properties (e.g., Lipinski’s rule of 5) using Molsoft ICM. Compounds were shipped at stock concentrations of 10 mM in DMSO and validated to be ≥85% purity using LC-MS at Mcule. The 94 compounds and two negative vehicle controls were randomized and blinded for the duration of the experiment and data analysis.

Vero cells (ATCC CCL-81) in 96-well plates were pre-treated with either DMSO or a 50 µM of the compound dissolved in DMSO for 1 h at 37 °C. Subsequently, the cells were infected with the recombinant RVFV MP-12 vaccine strain (rMP-12) at 0.2 MOI in the presence of the compound or DMSO. Following the replacement of the virus inoculum, the cells were further incubated for an additional 23 h at 37 °C with the presence of the 50 µM compound or DMSO. The virus titers in culture supernatants at 24 hpi (triplicate per compound) were determined through plaque assay using 12-well plates. We assessed a single dilution (1:1000) of the samples and recorded measurable virus titers in PFU/mL or classified them as either below or above the detection limit.

Three compounds (RGc-B05, RGc-F06, and RGc-F12) selected from this initial laboratory screening were further validated by the same pretreatment assay using Vero cells. In the control experiments, ribavirin and favipiravir, both at 25 µM, were also tested. The culture supernatants were screened for reductions in virus titers through a plaque assay. Identities of all supplied compounds were shared by Atomwise Inc. only after the initial screening process and data were returned to Atomwise, Inc.

### 2.3. Measurement of IC50 Values for Small Molecule Compounds

Three compounds (RGc-B05, RGc-F06, RGc-F12) were subjected to serial dilution and tested in a virus inhibition assay to determine their IC50 values. Vero cells were cultured in 24-well plates and pre-treated with either DMSO or the respective compound at 200, 100, 50, 25, 12.5, 6.25, or 3.13 µM. This incubation took place in a humidified CO_2_ incubator for 1 h at 37 °C. Following the pre-treatment, the cells were infected with rMP-12 virus at a MOI of 0.2 in the presence of the same concentration of compound for 1 h at 37 °C. Subsequently, the culture supernatants were replaced with fresh DMEM containing either DMSO or the compound, and the cells were further incubated at 37 °C for an additional 23 h. Virus titers in the culture supernatants at 24 hpi were measured in triplicate. Resulting virus titers (PFU/mL) were then converted to a percentage inhibition. Subsequently, the data were plotted using a non-linear regression model in GraphPad PRISM software (version 8.4.3) to determine IC50 values.

### 2.4. MTT Assay

The MTT assay was conducted to evaluate the cell toxicity of selected compounds. Vero cells were cultured in a 96-well plate and exposed to selected compounds at concentrations of 200, 100, 50, 25, 12.5, or 6.25 µM for 1 h at 37 °C. A control group with DMSO only was included. Following this treatment, the cells were either mock-infected or infected with rMP-12 at a MOI of 0.2, with the presence of the compound, for 1 h at 37 °C. Subsequently, the culture supernatants were replaced with fresh DMEM containing either DMSO or the compound. At 20 hpi, MTT reagent (MTT Cell Proliferation Kit I, Millipore Sigma, Burlington, MA, USA) was added into the culture supernatant, followed by a 4-h incubation at 37 °C. The resulting insoluble formazan crystals were dissolved using a solubilization solution, and the absorbance at 585 nm of colored solution was measured using the Accuris SmartReader 96 (Accuris Instruments, Edison, NJ, USA).

### 2.5. Northern Blot Analysis

Vero cells in 6-well plates were pre-treated with either DMSO or a selected compound at a concentration of 50 µM for 8 h at 37 °C. Subsequently, the cells were infected with rMP-12 or AMTV at 3 MOI in the presence of a 50 µM concentration of the compound or DMSO. After replacing the virus inoculum, the cells were further incubated for an additional 15 h at 37 °C with the presence of the 50 µM compound or DMSO. Virus titers in culture supernatants were measured at 16 hpi using a plaque assay. Total RNA was collected in Trizol reagent (Thermo Fisher Scientific) at 16 hpi and extracted RNA was subjected to Northern blot analysis as described previously using digoxigenin-labeled RNA probes specific to S-, M-, and L-segments of RVFV or those specific to AMTV S-segment [36,37].

### 2.6. In Vitro Efficacy of Compound Postexposure-Treatment in RVFV Infection

To assess antiviral efficacy under postexposure conditions, Vero cells infected with rMP-12 were treated with either DMSO or a selected compound at a concentration of 50 µM at 3 hpi. Virus titers were measured at 16 hpi using a plaque assay. Virion RNA was extracted from culture supernatants at 16 hpi following the previously described protocol [38]. Briefly, clarified culture supernatants were incubated with Benzonase (25U, Millipore Sigma) at 37 °C for 30 min, followed by cell lysis using Trizol LS (Life Technologies, Carlsbad, CA, USA). In each sample, 3 µg of in vitro synthesized chloramphenicol acetyltransferase RNA was added as a spike RNA. Viral RNA was then extracted using the Direct-zol RNA Miniprep Kit (Zymo Research, Irvine, CA, USA). First-stranded cDNA was synthesized from 480 ng of total RNA using the iScript Reverse Transcription Supermix (Bio-Rad Laboratories, Hercules, CA, USA). PCR reactions were prepared with 250 nM of each Taqman probe, 900 nM of each primer, ddPCR Supermix for Probes (Bio-Rad Laboratories), cDNA (corresponding to 1.2 ng RNA), and water (up to 25 μL). PCR cycling parameters were as follows: initial denaturation at 95 °C for 10 min, followed by 40 cycles of 94 °C for 30 s, 60 °C for 1 min, and a final denaturation step at 98 °C for 10 min. The number of droplets with positive and negative signals was measured using a Bio-Rad QX100 droplet reader. Data analysis was performed using QuantaSoft Version 1.4 (Bio-Rad Laboratories). The following primers and probes were used: S-segment RNA: forward primer (RV-NF: 5′-GGC TGG CTG GAC ATG C-3′), reverse primer (RV-NR: 5′-AGT GAC AGG AAG CCA CTC A-3′), and Taqman probe (Taq-RVFV-N: 5′FAM-CAG GCT TTG GTC GTC TTG AG-3′BHQ1) [39]. M-segment RNA detection: forward primer (RV-GnF: 5′-TCA CGA CAC CAT CAT TGC AA-3′), reverse primer (RV-GnR: 5′-GTT CCC ATG AGC ACT CAG AA-3′), and Taqman probe (Taq-RVFV-Gn: 5′HEX-AGG CTG ATC CAC CTA GCT GTG AC-3′BHQ1). L-segment RNA detection: forward primer (RV-LF: 5′-TGG AGC AGA TAG ACA ACC AGA-3′), reverse primer (RV-LR: 5′-CCT TAA GTG TGG CCA ACC TT-3′), and Taqman probe (Taq-RVFV-L: 5′HEX-TTC GAG AGC TCA GTG GGT TGA CT-3′BHQ1).

### 2.7. RT-qPCR Measurement of Viral RNA Synthesis

Vero cells in 24-well plates were pre-treated with either DMSO or a selected compound at a concentration of 50 µM for 8 h at 37 °C. Subsequently, the cells were adsorbed with rMP-12 at 3 MOI for 1 h on ice, in the presence of a 50 µM concentration of the compound or DMSO. After the 1-h incubation, the culture supernatant was replaced with warmed media containing the 50 µM compound or DMSO to initiate virus entry synchronously. Cells were lysed in Trizol at 0, 1, 2, and 4 hpi, and total RNA was extracted using the Direct-zol RNA Miniprep Kit (Zymo Research). For reverse transcription-quantitative PCR (RT-qPCR), first-stranded cDNA was synthesized from 200 ng of total RNA using the iScript Reverse Transcription Supermix (Bio-Rad Laboratories) following the manufacturer’s instructions. The PCR reaction with SsoAdvanced Universal Probes Supermix (Bio-Rad Laboratories) was performed using the Mic qPCR Cycler (4 channels): initial denaturation at 98 °C for 5 min, 40 cycles of 98 °C for 15 s, 60 °C for 45 s, and final denaturation at 98 °C for 10 min. The PCR reaction targeted the RVFV S-segment RNA and N mRNA with the forward primer (RV-NF), the reverse primer (RV-NR), and Taqman probe (Taq-RVFV-N), as described above. Serially 10-fold diluted S-segment RNA, which was in vitro synthesized from the pProT7-vS(+) plasmid using the MegaScript T7 Transcription kit (Thermo Fisher Scientific), were subjected to the 1st stranded RNA synthesis with iScript reverse transcriptase. Actual concentrations of RNA (copy number/µL in reaction) in each RNA dilution were measured by droplet digital PCR with the above-mentioned primers and probe set for RVFV S-segment, using the QX100 droplet generator and reader according to the manufacturer’s instructions. The resulting cDNA set derived from serial RNA dilutions with known RNA concentrations was used for the validation of the standard curve using the Mic qPCR.

In a similar manner, Vero cells were either mock-infected or infected with AMTV at a MOI of 3. The RT-qPCR was conducted as described above utilizing total RNA extracted at 16 hpi. For the detection of AMTV S-segment RNA and N mRNA, the following Taqman probe and primers were employed: Taqman probe, 5′-(HEX) TGC TGC AGG AGG AAT CCT AGA CC (BHQ1)-3′; forward primer, 5′-GTC TCA CAA CTA TCC ACG CG-3′; and reverse primer, 5′-GCA AGC ATC ACC TAG ACC TG-3′. The analysis of viral RNA copy number included the construction of a standard curve using diluted plasmid DNA encoding the full-length AMTV S-segment, namely pProT7-AMTV-S(+) [40].

### 2.8. In Vitro Efficacy of Compound Pre-Treatment in Bandavirus Infections

Vero cells were pre-treated with either DMSO or a specific compound at a concentration of 50 µM for 8 h at 37 °C. Subsequently, the cells were infected with HRTV or SFTSV at a MOI of 3, in the presence of a 50 µM concentration of the compound or DMSO. After replacing the virus inoculum, the cells were further incubated for an additional 23 h at 37 °C with the presence of the 50 µM compound or DMSO. Virus titers in culture supernatants were measured at 24 hpi using a focus-forming assay. In brief, Vero cells were infected with serially 10-fold diluted inoculum at 37 °C for 1 h. After removing the inoculum, cells were overlaid with 1xMEM containing 0.6% tragacanth gum, 5% FBS, 5% triphosphate broth, streptomycin, and penicillin. After 4 days post-infection, the overlay was removed, and cells were fixed with methanol and acetone (1:1). This was followed by immunostaining with anti-SFTSV N mouse monoclonal antibody (Clone 1A8, Immunology Consultants Laboratory Inc., Portland, OR, USA) or anti-HRTV N rabbit polyclonal antibody (ProSci Inc. Poway, CA, USA). The rabbit polyclonal antibody was generated through immunization with a peptide encoding amino acids 40–55 of the N protein (C-GLLRERGGENWRNDVK, with a cysteine added to the N terminus to facilitate conjugation to the carrier protein).

### 2.9. Statistical Analysis

Statistical analysis of research data was performed using GraphPad PRISM software (version 8.4.3). For group comparisons, statistical differences were assessed by one-way ANOVA followed by Tukey’s multiple comparison test, or two-way ANOVA followed by Sidak’s multiple comparison test. Likewise, for group comparisons involving viral titers or RNA copy numbers, ANOVA was applied to the arithmetic means of log_10_ values.

### 2.10. Ethics Statement

All experiments involving recombinant DNA and infectious RVFV, AMTV, HRTV, and SFTSV were conducted with the approval of the Institutional Biosafety Committee at UTMB, as specified in the Notification of Use (#2021017 and #2022004). All activities related to pathogenic HRTV and SFTSV took place in the GNL BSL-3 laboratory at UTMB.

## 3. Results

### 3.1. Identification of RVFV Replication Inhibitors Discovered through Virtual Screening

To identify potential binders of RVFV Gc, we conducted a vHTS that explored a library of over 8 million small-molecules interacting with the structural groove of RVFV Gc in the post-fusion conformation (PDB ID: 6EGT) (Figure 1a) [27]. From the library, 94 drug-like compounds (e.g., Lipinski’s rule of 5) were selected for in vitro screening. To validate the antiviral potency of the computer-selected compounds, we conducted a blinded screening of 94 selected small molecule compounds. This screening involved measuring the replication of the RVFV recombinant MP-12 strain (rMP-12), a vaccine strain classified as a Risk Group 2 non-select agent in the U.S.A. Notably, three compounds (RGc-B05, RGc-F06, and RGc-F12) consistently demonstrated a reduction in rMP-12 titers at 24 h post-infection in pretreated Vero cells, as depicted in Figure 1b. In comparison to the virus titers in the DMSO-treated group (control), the RGc-B05, RGc-F06, and RGc-F12 groups exhibited a 152, 95, or 24-fold reduction, respectively. Additionally, ribavirin or favipiravir demonstrated a 73 or 79-fold reduction in virus titers (*p* < 0.001, one-way ANOVA). Comparison of chemical identities revealed that RGc-B05, RGc-F06, and RGc-F12 share partial 2D structural similarity as illustrated in Figure 1c–e.

### 3.2. Antiviral Efficacy and Cell Toxicity of RGc-B05, RGc-F06, and RGc-F12 Compounds in Vero Cells

Next, we assessed the half-maximal inhibitory concentration (IC50) values for three chosen compounds across a concentration range of 3.1 to 200 µM. Vero cells were treated with either DMSO or the compounds for 1 h at 37 °C, followed by infection with the rMP-12 virus at a MOI of 2. IC50 values were determined based on virus titers at 24 hpi and are illustrated in Figure 2a–c. Notably, all three compounds exhibited dose-dependent virus inhibition, with IC50 values of 10.98 µM, 20.61 µM, and 11.90 µM for RGc-B05, RGc-F06, and RGc-F12, respectively. Furthermore, we evaluated the toxicity of these three compounds using the MTT assay, where toxicity was assessed by measuring the level of cell viability. As depicted in Figure 2d, cell viability from 20 to 24 hpi decreased by 8.6% and 6.3% with RGc-B05 and RGc-F06 at 200 µM, respectively. Meanwhile, treatment with RGc-F06 at 200 µM or DMSO alone did not reduce cell viability compared to the mock-treated control.

Furthermore, we examined the viability of rMP-12-infected cells following treatment with compounds (Figure 2e). This experiment was performed to evaluate the impact of compound treatments on the outcome of rMP-12-infected cells. Vero cells were treated with either DMSO or the compounds for 1 h at 37 °C, followed by infection with the rMP-12 virus at a MOI of 2. Cell viability was assessed from 20 to 24 h hpi using the MTT assay. At 200 µM, DMSO-treated rMP-12-infected cells exhibited a significant reduction in cell survival rate, reaching 62.5% compared to untreated rMP-12-infected cells. The treatment with RGc-F06 displayed a cell viability pattern resembling that of cells treated with DMSO alone, except under the 200 µM treatment condition, where RGc-F06 treatment resulted in 88.1% cell survival. Cells treated with RGc-B05 showed a 4.5% to 10.3% additional reduction in cell survival compared to the DMSO-treated group within the range of 6.3 to 100 µM, except for the 200 µM treatment condition, where RGc-B05 treatment led to 79.7% cell survival. Notably, treatment with RGc-F12 resulted in a significant reduction in cell survival of rMP-12-infected cells by 27.1% to 47.4% within the 6.3 to 25 µM range compared to DMSO-treated rMP-12-infected cells. However, RGc-F12-treated rMP-12-infected cells exhibited a better cell survival rate than the DMSO-treated group at 100 to 200 µM.

### 3.3. Potential Antiviral Mechanism of RGc-B05, RGc-F06, and RGc-F12 Compounds in Vero Cells

To understand the potential mechanism of virus inhibition by RGc-B05, RGc-F06, or RGc-F12 compounds, we used a one-step viral growth model using the rMP-12 virus. Vero cells were infected at 3 MOI, and virus RNA accumulation and titers were assessed at 16 hpi. Cells underwent pre-treatment with 50 µM of compounds or DMSO 8 h before infection, as opposed to 1 h, to optimize the antiviral effect under these stringent conditions. Upon pre-treatment with RGc-B05, RGc-F06, or RGc-F12 compounds, the accumulation of viral M- and L-segment RNA was nearly undetectable at 16 hpi, whereas a trace amount of S(−) RNA was observed in RGc-B05- or RGc-F06-treated cells (Figure 3a). Consistent with the result, the RGc-B05, RGc-F06, and RGc-F12 groups showed a 322, 216, or 189-fold reduction, respectively, compared to the DMSO-treated group (Figure 3b).

Next, we pre-treated Vero cells with either DMSO or 50 µM RGc-F06 for 8 h, followed by incubation with rMP-12 virus on ice for 1 h. After washing the cells with cold media, viral entry was synchronized by adding warmed media. Subsequently, we assessed viral RNA levels bound to Vero cells at 0 hpi and monitored viral RNA synthesis thereafter. Upon viral entry, rMP-12 virus can initiate primary transcription using viral L proteins derived from incoming virions, followed by the replication of genomic RNA [36]. Figure 3c illustrates that there were no significant differences in viral RNA bound to Vero cells at 0 hpi. However, in DMSO-treated cells, the viral S-segment RNA and N mRNA exhibited an increase from 2 hpi to 4 hpi. In contrast, the corresponding RNA levels in RGc-F06-treated cells were 27.2% and 13.3% at 2 and 4 hpi, respectively, compared to the DMSO-treated group. These findings suggest that the RGc-F06 treatment inhibited the process of viral RNA release during entry.

We also assessed whether these compounds exert antiviral effects after the establishment of viral infection. To investigate this, Vero cells were infected with rMP-12 virus at an MOI of 3, followed by treatment with 50 µM of compounds or DMSO 3 h post infection. However, the post-exposure treatment with RGc-B05, RGc-F06, or RGc-F12 compounds at 3 hpi did not significantly decrease virus titers at 16 hpi (Figure 3d). This result was further corroborated by examining RNA copy numbers of virion S-, M-, and L-segments in culture supernatants at 16 hpi (Figure 3e). The RNA copy numbers of virion samples from RGc-B05-, RGc-F06-, or RGc-F12-treated cells were comparable to those from the DMSO or untreated groups. This suggests that these compounds do not inhibit virus RNA replication or the assembly of viral RNA into virions but rather target the virus entry process.

### 3.4. Antiviral Efficacy of RGc-B05, RGc-F06, and RGc-F12 Compounds against Arumowot Virus or Bandaviruses

While the conformational structures surrounding the Gc fusion loop are conserved among phleboviruses and bandaviruses [27], it remained unclear whether small-molecule compounds specifically designed for RVFV exhibit antiviral activities against other phlebovirus or bandavirus species. Consequently, we conducted additional in vitro tests to assess the efficacy of RGc-B05, RGc-F06, and RGc-F12 compounds in Vero cells infected with AMTV, HRTV, or SFTSV. AMTV is a mosquito-borne phlebovirus genetically close to RVFV [43]. AMTV is non-pathogenic to human but maintains partial antigenic similarity with RVFV [44]. On the other hand, HRTV and SFTSV are tick-borne pathogenic bandaviruses that are less closely related to RVFV than AMTV [45]. Pre-treatment with RGc-B05, RGc-F06, or RGc-F12 compounds significantly suppressed AMTV replication in Vero cells, as illustrated in Figure 4a. However, the antiviral efficacy of RGc-B05, RGc-F06, or RGc-F12 against AMTV was not as robust as observed against RVFV, resulting in a 7.1-, 62.8-, or 42.3-fold inhibition compared to the DMSO-treated group. Meanwhile, treatment with RGc-B05, RGc-F06, or RGc-F12 led to a 2.8-, 9.4-, or 2.0-fold reduction in AMTV S genomic RNA and N mRNA, as determined by RT-qPCR (Figure 4b). Given the high viral input in the experiment, we also conducted Northern blot analysis to distinguish N mRNA and replicating AMTV S-segment RNA (Figure 4(c1)). The analysis revealed that the AMTV S-segment was barely detectable in cells treated with RGc-F06 at 16 hpi, whereas AMTV S-segment RNA and N mRNA were accumulated in cells treated with RGc-F12 pre-treatment, to a lesser extent with RGc-B05. In contrast, pre-treatment with 50 µM RGc-B05, RGc-F06, or RGc-F12 compounds did not reduce the replication of HRTV or SFTSV in Vero cells (Figure 4(c2,d)).

## 4. Discussion

We hypothesized that the existence of small molecule compounds capable of binding to a pocket formed around the fusion loop of the RVFV Gc protein could interfere with the structural arrangement during the fusion process. This interference could result in a reduction of ribonucleocapsid entry into the cytoplasm. This study utilized artificial intelligence-based technology to conduct virtual high throughput screening of small molecules targeting the post-fusion structure of the RVFV Gc protein, specifically in the vicinity of the groove binding to 1,2-dipropionyl-sn-glycero-3-phosphocholine (C3PC) adjacent to the fusion loop. This groove comprises residues C823, G824, C825, F826, N827, R776, C777, H778, L779, V780, D961, and P1135 [27]. Previous study showed that mutations such as H778A, L779A, F826S, F826N, N827A, and D961K could affect RVFV fusion activity [27,46,47,48]. Structural analysis further confirmed the significance of these residues along with R776 and V780, in the binding to the host membrane [27,49]. At a neutral pH, the binding of small molecules to the groove is impeded by the Gn protein [25]. Therefore, under neutral pH conditions, the binding of small molecules to Gc proteins is unlikely. Nevertheless, with the removal of the Gn shield in the late endosome at lower pH, small molecules gain access to the binding groove, potentially affecting the fusion activity. As a result of the virtual screening of small molecules, a total of 94 compounds were identified. Subsequent biological screening demonstrated that RGc-B05, RGc-F06, and RGc-F12 exhibit significant antiviral activity against the RVFV rMP-12 strain. Remarkably, all three hits among the 94 selected candidates successfully affected the entry pathway following viral attachment.

A fusion-inhibiting peptide, named RVFV-6, was previously described [26]. This 19-amino acid Gc stem peptide is suggested to interfere with the interaction between Gc domain III and the trimer core during stem zippering. Pretreatment of Vero E6 cells with the RVFV-6 peptide resulted in significant inhibition not only of RVFV but also of Andes virus and Ebola virus [26]. While small molecules targeting the fusion mechanism of RVFV have not been reported, our study revealed that three small molecules—RGc-B05, RGc-F06, and RGc-F12—possess potent antiviral activities in vitro when administered before virus infection. The consistent antiviral efficacy of these compounds was demonstrated after the pre-incubation of Vero cells before rMP-12 infection. These compounds showed no inhibitory effect on viral RNA replication or the packaging of viral genomic RNA when introduced after virus infection. Previous studies, however, have indicated that specific small molecule fusion inhibitors can exhibit therapeutic efficacy in animal models of respiratory syncytial virus or influenza virus infections [50,51]. Although susceptible rodents succumb rapidly to RVFV infection, future studies should explore the antiviral potency of small molecule fusion inhibitors in vivo. Alternatively, the concurrent use of a fusion inhibitor and a viral polymerase inhibitor could potentially result in additive or synergistic antiviral effects, as previously described [52].

Despite the limited amino acid homology between Gc proteins of RVFV and AMTV, the RGc-F06 compounds, initially designed for RVFV, exhibited notable antiviral activity against AMTV compared to RGc-B05 or RGc-F12. While pre-treatment with RGc-F12 reduced infectious virus titers of AMTV, this compound only weakly suppressed AMTV RNA replication. This suggests that the mechanism of AMTV inhibition by RGc-F12 may extend beyond the entry process. Interestingly, RGc-B05, RGc-F06, and RGc-F12 did not demonstrate any antiviral activities against HRTV and SFTSV. Despite the structural similarity in post-fusion structures between SFTSV Gc [53] and HRTV Gc [54], there appear to be structural differences in the binding groove structures between phleboviruses and bandaviruses, warranting further investigation. Nevertheless, the virtual screening approach using Gc groove structures holds promise for identifying broad-spectrum fusion inhibitors against bunyaviruses.

Viral fusion inhibitors, such as synthetic peptides, small molecules, and monoclonal antibodies [55], have been investigated as potent antivirals. However, there is limited research on antiviral candidates specifically targeting the fusion mechanism of bunyaviruses [26,56]. Leveraging artificial intelligence-based technology, our study efficiently identified small molecules capable of inhibiting the entry process of RVFV, which marks the initial milestone for further improvement of antiviral efficacy and safety for in vivo applications. Moving forward, additional pharmacological optimization of such small molecules and selection of optimal drug vehicles will help advance them toward pre-clinical and clinical testing.

## Figures and Tables

**Figure 1 viruses-16-00088-f001:**
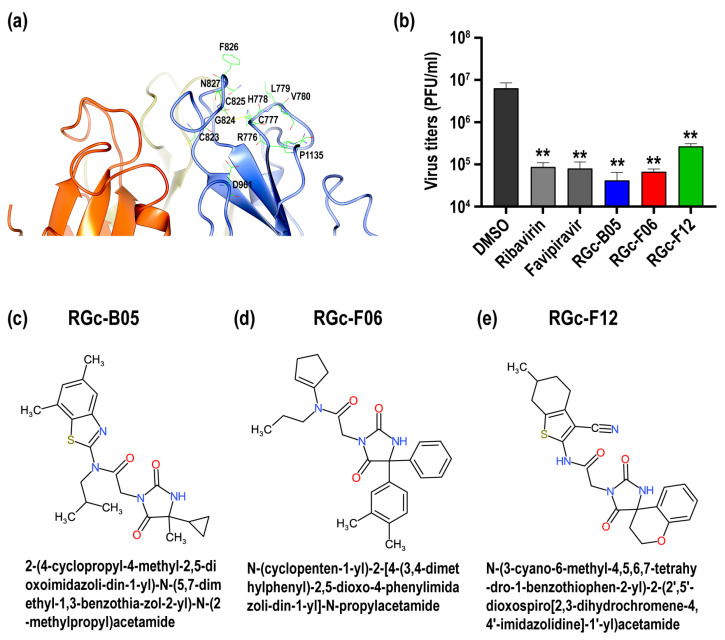
The screening of computer-selected small molecule compounds for inhibiting Rift Valley fever virus replication. (**a**) The virtual screening site on the RVFV Gc protein is highlighted within the post-fusion trimer (PDB ID: 6EGT), with labeled residues comprising the screening site. The illustration of the RVFV Gc structure was created using CCP4mg molecular graphics [41]. (**b**) Vero cells underwent mock treatment or were treated with either DMSO or a 50 µM of the compound dissolved in DMSO for 1 h at 37 °C. Subsequently, the cells were infected with the recombinant RVFV MP-12 strain (rMP-12) at 0.2 MOI in the presence of the compound or DMSO. Following media replacement, the cells were further incubated for 23 h with either 50 µM of the compound or DMSO. The graph illustrates the means ± standard deviations of rMP-12 titers from at least three independent experiments (one-way ANOVA; ** *p* < 0.001, vs. DMSO-treated control). (**c**–**e**) The 2D structures and IUPAC names for compound RGc-B05 (**c**), RGc-F06 (**d**), and RGc-F12 (**e**) are shown [42].

**Figure 2 viruses-16-00088-f002:**
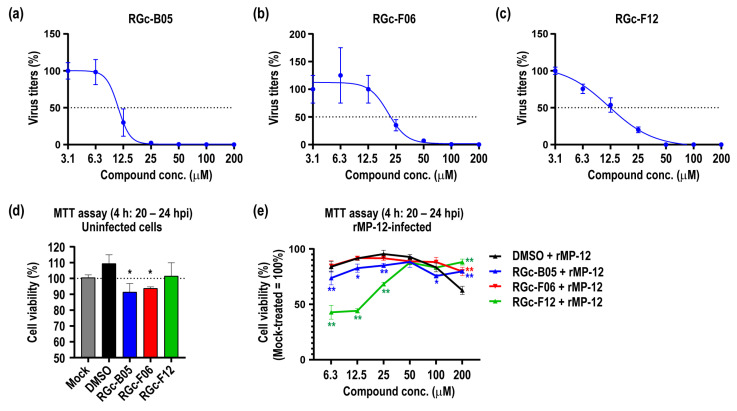
Viral inhibition and cell toxicity of three small molecule compounds in Vero cells. The half maximal inhibitory concentration (IC50) values for three selected compounds, (**a**) RGc-B05, (**b**) RGc-F06, and (**c**) RGc-F12, were depicted in graphs. The IC50 values were determined based on virus titers at 24 h post-infection (hpi) using Vero cells infected with the rMP-12 virus at a MOI of 0.2. Percentage of virus titers (PFU/mL) were plotted using a non-linear regression model. Dashed lines represent a 50% reduction in virus titers. Additionally, (**d**) cell viability (expressed as a percentage compared to mock-treated cells) of Vero cells treated with either DMSO or 200 µM of the compounds was assessed using the MTT assay. A dashed line represents the cell viability of mock-treated cells (100%). An asterisk denotes statistical significance determined by one-way ANOVA; * *p* < 0.05, compared to the DMSO-treated control. Furthermore, (**e**) cell viability (%) of Vero cells treated with DMSO or compounds, and subsequently infected with rMP-12 at an MOI of 0.2, was evaluated by the MTT assay. Data were normalized to mock-treated rMP-12-infected cells. An asterisk represents statistical significance determined by two-way ANOVA; * *p* < 0.05 or ** *p* < 0.01, compared to the DMSO-treated rMP-12-infected group.

**Figure 3 viruses-16-00088-f003:**
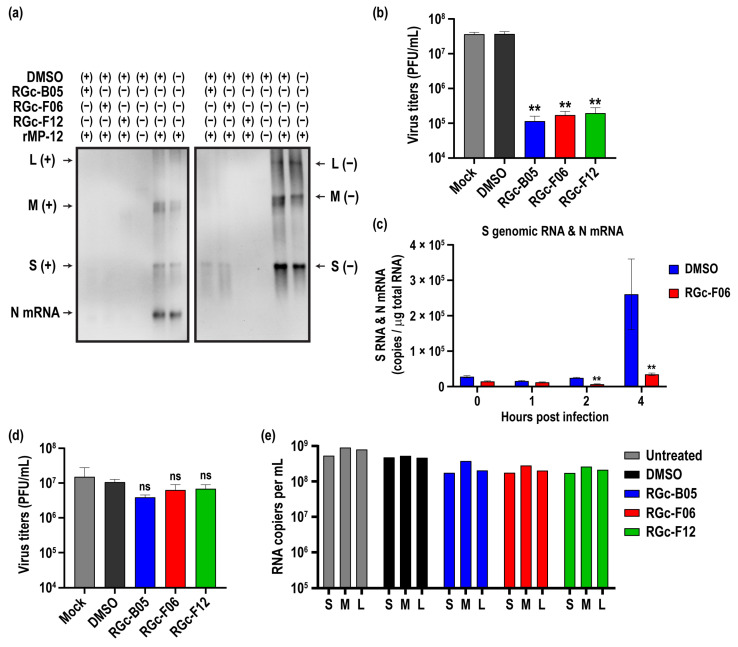
Inhibition of viral RNA synthesis by small molecule compounds in Vero cells. (**a**,**b**) Vero cells were pretreated with either mock treatment, 50 µM compounds, or DMSO 8 h prior to infection. After infection with the rMP-12 virus at a MOI of 3, total RNA and culture supernatants were collected at 16 h post-infection. These samples underwent Northern blot analysis (**a**) and virus titration using a plaque assay (**b**). The Northern blot analysis utilized a combination of RNA probes targeting N mRNA and positive-sense genomic S-, M-, and L-segment RNA [S(+), M(+), and L(+)] in the left panel, along with a mixture of RNA probes detecting negative-sense genomic S-, M-, and L-segment RNA [S(−), M(−), and L(−)] in the right panel. The graph (**b**) displays the mean ± standard deviations derived from at least three experiments. Asterisks indicate statistical significance determined by one-way ANOVA; ** *p* < 0.001, compared to the DMSO-treated group. (**c**) Vero cells were pre-treated with RGc-F06 compound at 50 µM or DMSO for 8 h, followed by rMP-12 infection at an MOI of 3 on ice for 1 h. Subsequently, cells were incubated with 37 °C warmed media to synchronize the viral entry process (defined as 0 hpi). Total RNA at 0, 1, 2, and 4 hpi underwent RT-qPCR using a Taqman probe detecting S-segment RNA and N mRNA. The graphs display the mean ± standard deviations derived from at least three experiments. Asterisks indicate statistical significance determined by two-way ANOVA; ** *p* < 0.001, compared to the DMSO-treated group. (**d**,**e**) Vero cells were either mock-treated or treated with 50 µM compounds or DMSO 3 h post infection. Following infection with the rMP-12 virus at a MOI of 3, total RNA and culture supernatants were collected at 16 h post-infection. The graph (**d**) presents the mean ± standard deviations of virus titers at 16 hpi derived from at least three experiments, with “ns” indicating “not significant” compared to the DMSO group. (**e**) The graph represents RNA copy numbers of S-, M-, and L-segment RNA of virions from culture supernatants collected at 16 hpi.

**Figure 4 viruses-16-00088-f004:**
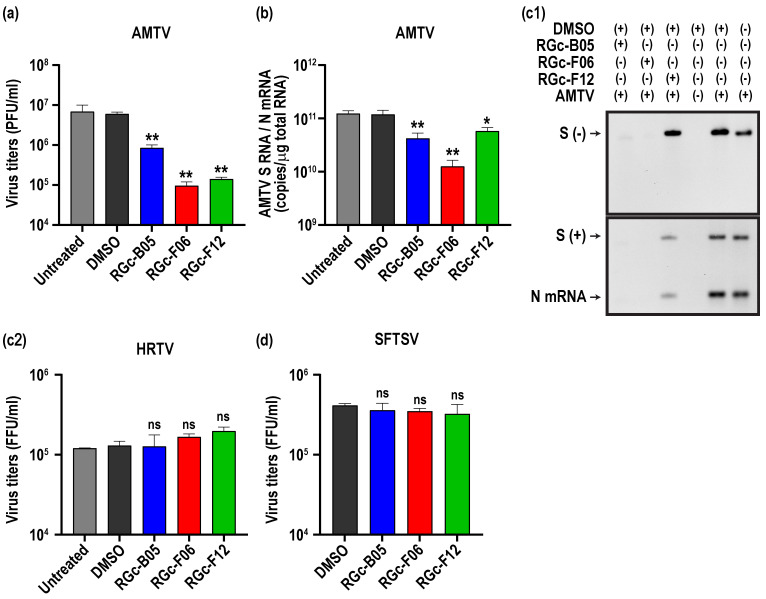
In vitro efficacy of small molecule compounds on Arumowot virus or bandaviruses. Vero cells were pretreated with mock treatment, 50 µM compounds, or DMSO 8 h before infection. Following infection with Arumowot virus (AMTV) at a MOI of 3, total RNA and culture supernatant were collected at 16 h post-infection (hpi). (**a**) Virus titers at 16 hpi were determined by the plaque assay. The graphs display the mean ± standard deviations from three experiments. Asterisks indicate statistical significance determined by one-way ANOVA; ** *p* < 0.001, compared to the DMSO-treated group. AMTV S RNA and N mRNA were detected by RT-qPCR (**b**) or Northern blotting (**c1**). The RT-qPCR utilized a Taqman probe detecting AMTV S-segment RNA and N mRNA. The graphs display the mean ± standard deviations derived from three experiments. Asterisks indicate statistical significance determined by one-way ANOVA; * *p* < 0.05, ** *p* < 0.01, compared to the DMSO-treated group. The Northern blot utilized the RNA probe detecting positive-sense genomic S-segment RNA and N mRNA of AMTV [S(+)], or the probe detecting negative-sense genomic S-segment RNA of AMTV [S(−)]. Similarly, after infection with Heartland virus MO-4 strain (HRTV) (**c2**) or Dabie bandavirus HB29 strain (SFTSV) (**d**) at a MOI of 3, culture supernatants were collected at 24 hpi to determine virus titers. The graphs present the mean ± standard deviations from three experiments, with “ns” indicating “not significant” compared to the DMSO.

## Data Availability

Data and materials are available through the agreement term made via the Office of Technology Transfer at UTMB.

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
