# Peer review of "Computer-Selected Antiviral Compounds: Assessing In Vitro Efficacies against Rift Valley Fever Virus"

_viruses, 2024, doi:10.3390/v16010088_

Round 1

Reviewer 1 Report

Comments and Suggestions for Authors

The authors describe a study conducted to evaluate the in-vitro efficacy of 3 chemical compounds to inhibit the replication of Rift Valley fever MP-12 virus in Vero cells. The approach was to screen chemical compounds using a structure-based artificial intelligence bioactivity predictor that identified million of small molecule compounds of which 3 out of 94 selected compounds on pretreatment of Vero cells inhibited the replication of RVFV MP-12 virus. However, as potential therapeutics, the compounds did not prevent viral RNA replication when administered three hours after infection. Also, the compounds inhibited Arumowot virus, another phlebovirus, but did not inhibit the tick-borne bandaviruses. Overall, the conclusion was that the study validated AI-based virtual high throughput screening as a rational approach for identifying antiviral candidates for Rift Valley fever virus and other bunyaviruses and that the virtual screening approach using Gc groove structures holds promise for identifying broad-spectrum fusion inhibitors against bunyaviruses. Also, additional pharmacological optimization of such small molecules will help to advance them toward pre-clinical and clinical testing. The rationale for this study is justified because there are no approved antiviral treatments for this Rift Valley fever disease in humans. Furthermore, the study is innovative because many small molecules have been identified as potent inhibitors of RVFV replication, but no small molecule have been reported that specifically interrupts the RVFV Gc fusion process to inhibit virus replication. While the rationale is clearly described and is based in part on the observations that many small molecules have been identified as potent inhibitors of RVFV replication, the rationale could be improved by including a brief summary of the results of evaluating these small molecules as potential antivirals for the inhibition of RVFV replication in-vitro and/or in-vivo. Overall, the contents of this manuscript are readily understood and the approach, and methods are appropriate for achieving the results. The results present an excellent analysis and description of the observations, and the discussion further clarifies the observations and addresses the observations in relation to relevant findings by others. The discussion could be strengthene by commenting on the observation that the compounds did not exibit any inhibitory effect on viral RNA replication or the packaging of viral genomic RNA when introduced after virus infection, and therefore, did not demonstrate any therapeutic efficacy. Furthermore, can a therapeutic be identified and used effectively for Rift valley fever, a virus that causes an acute viral infection that is likely to be much to rapid to be inhibited by administering therapeutics. Another minor comment pertains to the use of Dimethyl Sulfoxide (DMSO) is if the development of the candidate antivirals are extended to evaluation in animals, will this chemical be have any adverse health consequences?   

Author Response

  1. While the rationale is clearly described and is based in part on the observations that many small molecules have been identified as potent inhibitors of RVFV replication, the rationale could be improved by including a brief summary of the results of evaluating these small molecules as potential antivirals for the inhibition of RVFV replication in-vitro and/or in-vivo. The discussion could be strengthened by commenting on the observation that the compounds did not exhibit any inhibitory effect on viral RNA replication or the packaging of viral genomic RNA when introduced after virus infection, and therefore, did not demonstrate any therapeutic efficacy. Furthermore, can a therapeutic be identified and used effectively for Rift valley fever, a virus that causes an acute viral infection that is likely to be much to rapid to be inhibited by administering therapeutics.

Response: Thank you for your suggestion. We have incorporated an additional paragraph describing the potential in vivo antiviral potency, comparing it to published RVFV fusion inhibitors, as follows:

“A fusion-inhibiting peptide, named RVFV-6, was previously described [26]. This 19-amino acid Gc stem peptide is suggested to interfere with the interaction between Gc domain III and the trimer core during stem zippering. Pretreatment of Vero E6 cells with the RVFV-6 peptide resulted in significant inhibition not only of RVFV but also of Andes virus and Ebola virus [26]. While small molecules targeting the fusion mechanism of RVFV have not been reported, our study revealed that three small molecules—RGc-B05, RGc-F06, and RGc-F12—possess potent antiviral activities in vitro when administered before virus infection. The consistent antiviral efficacy of these compounds was demonstrated after the pre-incubation of Vero cells before rMP-12 infection. These compounds showed no inhibitory effect on viral RNA replication or the packaging of viral genomic RNA when introduced after virus infection. Previous studies, however, have indicated that specific small molecule fusion inhibitors can exhibit therapeutic efficacy in animal models of respiratory syncytial virus or influenza virus infections [50, 51]. Although susceptible rodents succumb rapidly to RVFV infection, future studies should explore the antiviral potency of small molecule fusion inhibitors in vivo. Alternatively, the concurrent use of a fusion inhibitor and a viral polymerase inhibitor could potentially result in additive or synergistic antiviral effects, as previously described [52].”

  1. Another minor comment pertains to the use of Dimethyl Sulfoxide (DMSO) is if the development of the candidate antivirals are extended to evaluation in animals, will this chemical be have any adverse health consequences?   

Response: Thank you for the suggestion. We used DMSO as a solvent for small molecule for our in vitro study. However, we will need to select optimal drug vehicles to be used in vivo. Following revised sentence was included:

“Moving forward, additional pharmacological optimization of such small molecules and selection of optimal drug vehicles will help advance them toward pre-clinical and clinical testing.”

Reviewer 2 Report

Comments and Suggestions for Authors

The manuscript by Alkan and colleagues describes a straightforward screen for compounds that can inhibit RVFV entry. Virtual screening was used to identify 94 potential compounds from a large chemical library that might be able to interact with a structural groove in the viral Gc protein. Screening in cells revealed antiviral activity for 3 compounds, 1 of which was clearly better in terms of both antiviral activity and reduced toxicity. These compounds were also tested against three other bunyaviruses, AMTV, HRTV and SFTSV. Interestingly, the compound that was most effective against RVFV was not the most effective against AMTV. None of the compounds were effective against the more distantly related HRTV and SFTSV.  This is a well-done study with appropriate experimental design and conclusions.  It is also a well written manuscript.  I have only one very minor comment, which is that the species name Dabie bandavirus should not be used to refer to the actual virus used in the study.  A species is a human concept; it is not a physical entity, but only a name for a collection of related virus strains.  A species cannot infect cells or be assayed, since it does not exist as a physical entity. I am not in the bunyavirus field but it appears that the field needs to sort out what name to use for this virus when referring to virus actually used in experiments or detected in nature, and not the species.

Author Response

  1. I have only one very minor comment, which is that the species name Dabie bandavirus should not be used to refer to the actual virus used in the study.  A species is a human concept; it is not a physical entity, but only a name for a collection of related virus strains.  A species cannot infect cells or be assayed, since it does not exist as a physical entity. I am not in the bunyavirus field but it appears that the field needs to sort out what name to use for this virus when referring to virus actually used in experiments or detected in nature, and not the species.

Response: Thank you for your suggestion. We have incorporated the correction in the revised manuscript by using the term "bandaviruses" to encompass both HRTV and SFTSV. Additionally, we referred to SFTSV as "Dabie bandavirus HB29 strain."